# Novel methods for estimating the instantaneous and overall COVID-19 case fatality risk among care home residents in England

Christopher E. Overton[1,2,3,4,5]*, Luke Webb[1], Uma Datta[6], Mike Fursman[6], Jo Hardstaff[7], Iina Hiironen[7], Karthik Paranthaman[7], Heather Riley[1], James Sedgwick[7], Julia Verne[8,9], Steve Willner[8], Lorenzo Pellis[1,3,4,11], Ian Hall[1,2,3,10,11]

1 Department of Mathematics, University of Manchester, Manchester, United Kingdom, 2 Clinical data science unit, Manchester University NHS Foundation Trust, Manchester, United Kingdom, 3 Joint UNIversities Pandemic and Epidemiological Research, https://maths.org/juniper/, Cambridge, United Kingdom, 4 Department of Mathematical Sciences, University of Liverpool, Liverpool, United Kingdom, 5 Data, Analytics and Surveillance, UK Health Security Agency, London, United Kingdom, 6 Care Quality Commission, London, United Kingdom, 7 Field Service, National Infection Service, Public Health England, London, United Kingdom, 8 Adult Social Care Team, Public Health England, London, United Kingdom, 9 Office for Health Improvement and Disparities, Department of Health and Social Care, London, United Kingdom, 10 Emergency Preparedness, Health Protection Division, Public Health England, London, United Kingdom, 11 Alan Turing Institute, London, United Kingdom

☯ These authors contributed equally to this work.

* christopher.overton@manchester.ac.uk

## Abstract

The COVID-19 pandemic has had high mortality rates in the elderly and frail worldwide, particularly in care homes. This is driven by the difficulty of isolating care homes from the wider community, the large population sizes within care facilities (relative to typical households), and the age/frailty of the residents. To quantify the mortality risk posed by disease, the case fatality risk (CFR) is an important tool. This quantifies the proportion of cases that result in death. Throughout the pandemic, CFR amongst care home residents in England has been monitored closely. To estimate CFR, we apply both novel and existing methods to data on deaths in care homes, collected by Public Health England and the Care Quality Commission. We compare these different methods, evaluating their relative strengths and weaknesses. Using these methods, we estimate temporal trends in the instantaneous CFR (at both daily and weekly resolutions) and the overall CFR across the whole of England, and dis-aggregated at regional level. We also investigate how the CFR varies based on age and on the type of care required, dis-aggregating by whether care homes include nursing staff and by age of residents. This work has contributed to the summary of measures used for monitoring the UK epidemic.

**Data Availability Statement:** Codes for estimating the case fatality ratio are provided at: https://github.com/OvertonC/CFR_toolbox. These include the

scripts designed for handling the care home data as well as general scripts, which any aggregated time series and delay data can be loaded into. The data used in this analysis was provided through a data sharing agreement with Public Health England (now UK Health Security Agency) as part of the urgent response to the COVID-19 outbreak, so unfortunately cannot be made available by the authors. An application for data access can be made to the UK Health Security Agency. Data requests can be made to the Office for Data Release (https://www.gov.uk/government/publications/accessing-ukhsa-protected-data) and contacting DateAccess@ukhsa.gov.uk. All requests to access data are reviewed by the Office for Data Release and are subject to strict confidentiality provisions in line with the requirements of: the common law duty of confidentiality, data protection legislation (including the General Data Protection Regulation, 8 Caldicott principles, the Information Commissioner's statutory data sharing code of practice, the national data opt-out programme.

**Funding:** L.P. and C.E.O. are funded by the Wellcome Trust and the Royal Society (grant no. 202562/Z/16/Z). I.H. is supported by the National Institute for Health Research Health Protection Research Unit (NIHR HPRU) in Emergency Preparedness and Response and the National Institute for Health Research Policy Research Programme in Operational Research (OPERA). CEO, LP, and IH are supported by the UKRI through the JUNIPER modelling consortium [grant number MR/V038613/1]. The funders had no role in study design, data collection and analysis, decision to publish, or preparation of the manuscript.

**Competing interests:** The authors have declared that no competing interests exist.

## Author summary

During an epidemic, the case fatality risk (CFR), i.e. the probability that an individual dies after testing positive for a disease, is a key parameter informing the public health response. However, calculating the CFR is not trivial, since there are cases who may die in the future but have not died yet. Therefore, statistical methods are required to correct for the distribution of times between testing positive and dying. In this paper, we derive multiple methods, some existing and some novel, within a consistent methodological framework. This allows us to understand how these different approaches are related and their relative strengths and weaknesses. During the COVID-19 pandemic, care homes have been particularly affected, due to the high risk of COVID-19-associated mortality in the frail and elderly. We apply our CFR methods to data from English care homes to analyse changes in the care home CFR throughout the pandemic.

## Introduction

Since December 2019, COVID-19 has spread rapidly throughout the world, leading to a global pandemic and public health crises in many countries, including the United Kingdom. In England, as in many other countries worldwide, care homes/residential homes for the elderly and frail have been particularly badly affected. For example, between 4 March 2020 and 7 August 2020 there were an estimated 29,500 excess deaths in care homes in England when compared to the same period in previous years [1], many of which are likely to be due to COVID-19. Care home populations are typically elderly, and numerous studies have identified an increasing risk of severe disease outcomes as age increases in patients with COVID-19 [2–5]. In addition, there is constant contact with the outside community, as new residents are admitted and as care home staff travel daily between their home circumstances and the care home, which makes is challenging to protect care home residents from outbreaks in the wider community.

Throughout the COVID-19 pandemic, easy-to-interpret metrics have garnered much attention due to the need for rapid dissemination of information to inform policy and the interest from the general public. The case fatality risk (CFR, sometimes called case fatality rate or ratio) is one such measure, and can be a powerful tool when quantifying the risk of adverse outcomes. There are different ways to interpret the CFR, but all fundamentally ask the same question: what is the risk of dying from/with a disease? In this paper, we use the convention that the CFR refers to the proportion of confirmed cases that lead to death and refer to the true death rate of the disease as the infection fatality ratio (IFR), i.e. the proportion of deaths out of all infections, detected or not. Ideally the CFR provides a close estimate of the IFR. However, this is not always the case in practice, as case ascertainment may be low or vary across geographies depending on access to testing. In particular, preferential ascertainment of severe cases can lead to CFR being biased relative to the IHR [6–8]. This is particularly true in relation to COVID-19, owing to the large proportion of asymptomatic and mildly symptomatic cases which could go undetected [9]. Some work has been done to look at the IFR for COVID-19 through attempts to infer case-ascertainment rate [2, 10]. In this paper though, we focus specifically on methods to compute the CFR, which is potentially more useful, for example in forecasting death rates, as we know the number of cases rather than the number of infections.

Estimating the CFR during an ongoing epidemic, or indeed pandemic, can be challenging, in part due to the delay between infection and outcome. At the time of computation, many cases may have not yet reached their final outcomes. This right-censoring means that simply

dividing the number of deaths by the number of cases can give misleading results. This is particularly true in the early stages of an epidemic, as these cases with unknown outcomes represent a larger proportion of the overall sample. There are various methods to address these biases, but in all methods, it is important to account for the distribution of delays between infection and outcome [11, 12]. Furthermore, an important and often overlooked consideration is that we may see changes in the CFR over time due to, for example, changes in testing rates, treatments, or severity of the disease through mutation. Therefore, it may be insufficient to consider a constant or overall CFR, and the instantaneous CFR, which looks at the CFR for each day of the epidemic, may be more useful.

If data are available linking cases to deaths, then the CFR is straightforward to calculate by following all positive individuals forward in time. However, often such data are not available. During the early stages of this project, the only available deaths data for care home residents were collected by the Care Quality Commission (CQC), and cases data were collected by Public Health England (PHE). These data could not be linked. Therefore, statistical methods for estimating the CFR were required.

A number of statistical methods have been used to compute the CFR for COVID-19 at different stages of the pandemic, amongst different populations and demographics. As such, reported estimates for the CFR of COVID-19 have varied wildly, with values as low as 0.15% [13] and as high as 25.9% [5]. Many reports use the measure of total deaths divided by total cases to calculate an overall CFR. These include initial reports from Wuhan, which reported a variety of estimates for the CFR, as high as 5.25% in Wuhan [14] itself, but lower in the rest of mainland China at 0.15% [14] or 0.4% [13]. However, when accounting for the censoring of outcomes, estimates were significantly higher, at 12.2% in Wuhan and 0.9% in the rest of mainland China [15]. Furthermore, analysis of outbreaks in other countries has also found that the crude CFR under-estimates the delay-adjusted CFR [5, 16, 17] (for the overall CFR). The crude instantaneous CFR depends on the phase of the epidemic, and is underestimated during growth phases and overestimated during decay phases.

In this paper, we investigate three approaches in application to data from care homes in England: cohort method, backward method, and forward method. The cohort method is perhaps the most straightforward, but is heavily affected by the right-censoring and requires very high resolution data. The backward method is widely used in the literature [2, 10], since it requires relatively simple data and is not sensitive to the right-censoring. Forward methods have been used [18], but are less common than the other two. Starting from the cohort method, we derive a novel forward method. The resulting model is similar to the existing forward method of Reich et al. (2012) [18]. By investigating these different approaches, we compare the CFR estimates obtained and discuss their relative strengths and weaknesses.

When estimating the CFR, there are two general perspectives considered: overall CFR and instantaneous CFR. The overall CFR describes the proportion of cases to date that resulted in death, whereas the instantaneous CFR describes the proportion of cases within the given time frame that resulted in death. The overall CFR therefore paints a picture of total mortality during the pandemic. The instantaneous CFR on the other hand paints a picture of how mortality risk changes over time. The existing instantaneous methods describe daily instantaneous CFR, which is the CFR on a given day. This provides a high-resolution temporal trend, but can be very sensitive to small numbers of observations. To address this, we derive novel weekly instantaneous methods, which describe the CFR in a given week. By aggregating across weeks, we ensure a larger number of observations, whilst maintaining decent temporal resolution. Through this real-time quantification of mortality risk, it is possible to identify changes in mortality risk over time, for example due to the emergence of new variants [19] or different epidemic trends across geographic regions. Additionally, the instantaneous CFR is essential

for constructing forecasting models for deaths. Such forecasts cannot be parameterised using the overall CFR, since this would not capture temporal changes in the relationship between cases and deaths.

In addition to temporal trends across England, we also look at regional heterogeneity in the CFR, looking at different NHS regions across England. Some of the potential drivers of regional heterogeneity are differences between regions in demographics [20], care availability [21], epidemic trajectories, testing rates, and the emergence of novel variants. By investigating regional data, we can provide region specific estimates for the CFR. We also compare two different data sources for the deaths data: PHE 28-day care home deaths, by date of death, and CQC confirmed care home deaths, by date of report. These different data sets introduce distinct censoring issues. The PHE data are explicitly censored 28 days after a positive test, whereas CQC deaths can include any deaths where COVID-19 has contributed, and can also be reported any length of time after the death occurs. This can lead to unclear right-censoring issues with the CQC data.

The methods used and developed in this paper have been applied throughout the COVID-19 pandemic to provide real-time updates on the case fatality risk in English care homes. These have fed into the PHE Joint Modelling Team weekly reports, the SAGE Social Care Working Group, and the Cabinet Office COVID-19 dashboard. The main focus within these reports has been monitoring the impact of the vaccination programme on mortality in care homes, as well as identifying other potential changes in mortality, related to, for example, new variants. Whilst these methods are useful for investigating temporal trends in the CFR, they do not directly provide causal explanations for changes therein. Use of these CFR time series within more complicated statistical frameworks may allow for such explanations to be extracted from the data.

## 1 Data

To investigate the CFR in care homes, we use two data sources. Firstly, from PHE systems, we extracted non-identifiable, time series data on persons with confirmed SARS-CoV-2 infection and resident in care homes in England. Residence in a care home was identified by matching the full residential addresses of patients against three reference databases—Ordnance Survey (OS), Care Quality Commission (CQC) list of registered LTCFs and OS AddressBase Premium database. Cases not matched through the above process were manually matched by NHS number to the Master Patient Index held by NHS England. Dates of death for care home residents who died after a positive COVID-19 test cases were identified using a combination of methods previously described [22]. From this dataset, we extracted COVID-19 deaths through the PHE definition, using deaths within 28 days of the positive test. To increase reliability of identification of persons resident in care homes, we included only those aged over 65 years in this analysis. This, any care home residents under 65 are excluded, and this analysis should be interpreted as the CFR for care home residents over 65 years of age.

The second data set we use was extracted from CQC systems (full descriptions of the data can be found in S1 Text). This data set records the number of COVID-19 deaths in each English care home. All care homes are required to report deaths to CQC, with the deaths disaggregated by type of death (e.g. Confirmed COVID, Suspected COVID, not COVID), and by place of death (e.g. hospital, place of residence). The confirmed deaths in this data set will be deaths that care home staff are confident COVID-19 has contributed towards, such as based on their doctor's notes. In our analyses, we only include deaths that have been reported as confirmed, and include all places of death. During the first wave, a high proportion of deaths were reported as suspected, due to low rates of testing. These are not included in the analysis, which

results in deaths being under-counted during the first wave. However, since the testing rate was so low during the first wave, the CFR estimates here are very unreliable, regardless of the inclusion/exclusion of suspected COVID-19 deaths. Since this is reported data, we do not have the number of deaths by date of death but instead the date these deaths were reported to CQC. This adds an extra lag to the data, since it may take a few days for the deaths to be reported, and also introduces extra biases, such as a weekend effect where fewer deaths are reported on Saturdays and Sundays, which are then reported on the following Monday/Tuesday. Generally, the lag time between death and reporting is very short, taking a few days, but there is potentially a heavy tail on the distribution of delays.

By linking the two data sets by postcode, we can obtain time series for the number of positive tests per day and the number of deaths (and deaths reported) per day. At England level, these three time series are plotted in Fig 1. Positive tests and deaths both follow a similar shape, though there is a temporal offset, due to the delay between testing positive and dying, and scaling offset, due to only a proportion of tests resulting in deaths. Access to testing was limited in the first wave (until May 2020). Over summer 2020, testing capacity grew rapidly, becoming relatively stable for the subsequent waves. Comparing the two deaths time series, we see that both follow the same general trend. We can see weekend effects in both in the PHE positive test data and CQC deaths data, but not in the PHE death data. This is expected, as death reporting to CQC and testing behaviour is heavily biased by the day of week, whereas date of death is not.

Individuals in English care homes are substantially older, on average, than those in the wider population. Among care home residents who tested positive for COVID-19 in our data, the median age was 86, with 5th percentile equal to 69 and 95th percentile equal to 97. This elderly age profile leads to substantial COVID-19 mortality risk, relative to the wider population.

## 2 Methods

In this section, we describe different methods for calculating the case fatality risk, and their relative strengths and weaknesses. Firstly, we describe some commonly applied methods. We then develop novel methods that build on these, leading to a variety of techniques that can be used to estimate the CFR. The aim of all these methods is to account for the delay between infection and death. We cannot simply divide the number of deaths on a given day by the number of cases on that day, because many deaths are likely to be in individuals that tested positive on prior days. If the epidemic were constant, this would not be an issue, as the number of cases would be the same each day. However, in reality epidemics change over time, with cases either growing or decaying. Therefore, it is important to construct an appropriate denominator that considers the number of cases on historic days and scales by the time to death delay distribution. We present three methods for estimating the CFR that account for these delays, which we refer to as "cohort", "backward", and "forward". If linked data are available, such as the PHE data, the cohort method can be applied to easily calculate the CFR. In the absence of linked data, the statistical backward and forward methods are necessary.

When estimating the CFR, there are different resolutions that can be considered. These fall into two general categories, which we refer to as "instantaneous" and "overall". The instantaneous measure describes the CFR within a given time frame, whereas the overall measure describes the CFR up to a given point in time. Therefore, the instantaneous CFR can be interpreted as the mortality risk within the corresponding time frame, which could be a daily CFR or a weekly CFR, for example, corresponding to the mortality risk on that day or during that week, respectively. The overall CFR on the other hand can be interpreted as the mortality risk

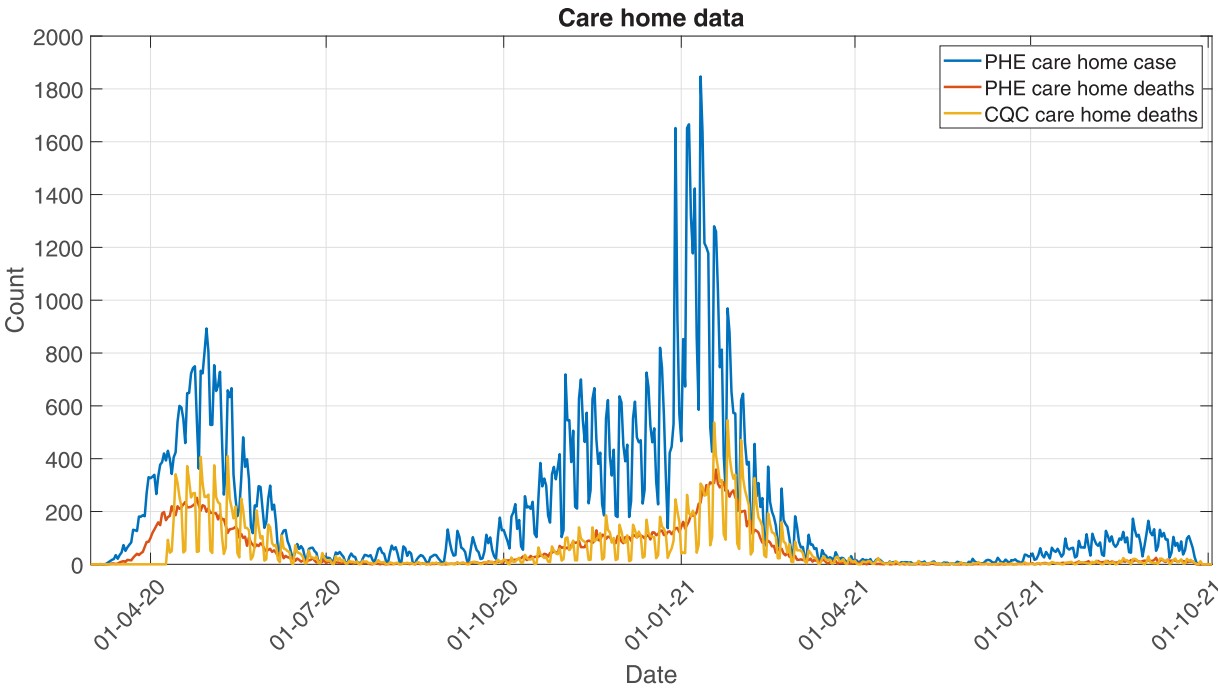

**Fig 1. Plotting the time series for the 3 data streams: Cases, PHE deaths, and CQC deaths.**

of all cases up to and including the given day. This does not provide a measure of real-time risk, since the risk may change during the pandemic, but rather a measure of the mortality impact of the whole pandemic. Therefore, instantaneous measures are more useful for assessing temporal trends in the CFR and quantifying real-time risk, whereas the overall measures are more useful for quantifying how the target population has been affected.

In this paper, we adopt different notations for each type of CFR. The daily-instantaneous CFR is denoted $CFR(t)$, the weekly-instantaneous CFR is denoted $wCFR(t)$, and the overall CFR is denoted $oCFR(t)$. Within each of these notations, we identify the distinct estimation methods: cohort, forward, and backward. These are denoted by a subscript $_{\text{cohort}}$, $_f$, and $_b$, respectively. For example, the forward daily-instantaneous CFR is denoted $CFR_f(t)$ and the cohort weekly-instantaneous CFR is denoted $wCFR_{\text{cohort}}(t)$.

## 2.1 Existing methods for calculating the CFR

**2.1.1 Cohort methods.** Perhaps the most intuitive approach for estimating the case fatality risk is the cohort method. In this method, to estimate the CFR on a given day (let us call this day $\tau$), one needs to identify all individuals that became a case on this day. Following these cases forward in time, it is possible to count the number of deaths observed within this group, and the CFR is given by the number of deaths at $t = \infty$ divided by the number of cases on $t = \tau$. This approach is easy to calculate, and the resultant CFR is easy to interpret, since it reflects the proportion of cases on day $\tau$ that result in death. However, there are some drawbacks. Firstly, and perhaps most severely, this method requires individual-level data linking cases to deaths. Such data may be challenging to access, with often only aggregated time series data available. Secondly, this method is subject to right-censoring, since many individuals may not have experienced their final outcomes (death or recovery). Therefore, whilst estimates of the CFR where $\tau$ is sufficiently long ago may be reliable, since we can assume most deaths will

have occurred, as $\tau$ approaches the current date, the estimates will be more biased by the right-censoring. To account for this issue, we propose two potential solutions. Firstly, using data on the distribution of delays from becoming a case to dying, a threshold can be constructed based on the 99% prediction interval of this distribution, which we will denote $t_{max}$. Using this, we can truncate the CFR time series, so that the most recent estimate considered reliable is when $\tau = T - t_{max}$, where $T$ is the last day of data. Another solution arises from the definition of deaths considered. One metric for measuring COVID-19 deaths is the PHE 28-day death definition. Under this definition, if an individual has not died within 28 days of their first positive test, they are assumed to have recovered. Therefore, this applies a hard cutoff to the maximum time between becoming a test and dying. Using this cutoff, the CFR time series can be truncated so that the most recent estimate that is considered reliable is when $\tau = T - 28$. This is the cutoff we use throughout the analyses considered in this paper.

This describes how to generate daily-instantaneous CFR estimates (which describe the proportion of cases on a given day that result in death) using the cohort method. Weekly CFR estimates (which describes the proportion of cases in a given week that result in death) can be similarly obtained, by looking at the outcomes for all individuals who test positive within a given week rather than a given day. Similarly, overall CFR estimates can be generated by looking at the outcomes of all individuals that have tested positive up to and including a given date.

**2.1.2 Backward methods.** When individual-level linked data are not available, the CFR has to be estimated from time series data, making the cohort method inapplicable. Perhaps the most common approach for estimating the CFR from time series data considers the number of deaths on a given day, and then constructs an appropriate denominator by looking at historic cases data [2]. We will refer to this method as the backward method, denoted by a subscript $b$ (e.g. $CFR_b(t)$), since it involves looking backward in time from the date of death when constructing the denominator. This denominator is calculated by looking at the number of cases on each previous day and scaling this by the proportion of deaths that would occur on day $t$, conditional on death. This gives the daily-instantaneous CFR under the backward method as (See S3 Text):

$$CFR_b(t) = \frac{d(t)}{\sum_{x=0}^{t} C(x)g(t-x|x)},$$

(1)

where $d(t)$ is the number of deaths on day $t$, $C(x)$ is the number of cases on day $x$ and $g(y|x)$ is the probability that an individual dies $y$ days after testing positive, given that they test positive on day $x$.

In addition to the daily-instantaneous method, there exist methods for calculating the overall CFR using the backward method [2]. The overall backward CFR on day $t$ is given by the number of deaths up to and including day $t$ divided by the expected number of cases that could have ended in death on or before day $t$. That is,

$$oCFR_b(t) = \frac{\sum_{x=0}^{t} d(x)}{\sum_{x=0}^{t} c(x)F_\theta(t-x|x)},$$

(2)

where $F_\theta(y|x)$ is the cumulative distribution function of the testing to death delay, given that the individual tests positive on day $x$.

The main benefit of this backward approach is that, once a delay distribution from testing positive to death has been estimated, this method only requires time series data rather than an individual-level line-list. Such data are often readily available, so this method may be easier to implement in practice. However, since this method requires adjusting for the death delay, we need some line-list data from which to extract the delay distribution (or delay estimates to exist in the literature). This does not need to be complete, as long as the estimates are representative of the full cohort, so is easier to obtain than full individual-level data. Another benefit is that the backward method is not subject to right-censoring, and CFR estimates can be calculated up until the final day of the available data (subject to any back-filling issues due to reporting delays). This is because this approach looks backward to see how historic cases contributed to the deaths today, so is not dependent on future outcomes.

A downside of this backward method is that the results are harder to interpret than the cohort method. In the latter, the daily CFR indicates the proportion of cases on the given day that are expected to die. The backward method instead indicates the proportion of cases that led to deaths on the given day. This means that the CFR from the backward method is less useful for parameterising deaths forecasting tools, since it does not link the cases data to deaths (instead linking deaths to cases). However, it still provides a reliable indicator of temporal trends in mortality risk, which is perhaps the main purpose of the CFR.

## 2.2 Novel methods for calculating the CFR

Motivated by the interpretability of the cohort method, we derive on approximation to this method that uses time series data rather than line-list, which we refer to as the forward method. This approach will also require delay distribution data, so requires the same inputs as the backward method, making them comparable in their applicability. The resulting model is similar to the model developed by Reich et al. (2012) [18] for calculating the infection fatality ratio across covariate groups with varying reporting rates. We derive both instantaneous and overall forward methods.

Whilst the daily instantaneous CFR provides a high-resolution temporal trend, the data may not be sufficient for reliable estimation of the daily CFR, since the number of daily observations can be very low. This does not present a substantial issue when estimating the CFR in England, however, when disaggregating the data by region, age, or other potential covariates, the sample sizes become much smaller. To address this, we develop novel methods for estimating the weekly-instantaneous CFR, using both forward and backward methods. By aggregating the instantaneous CFR into a weekly rather than daily time-window, we can generate reliable estimates for the CFR when numbers are low. Such methods may be more applicable than daily estimates for smaller geographies, such as the devolved administrations of Scotland, Wales, and Northern Ireland, since smaller geographies are likely to have fewer cases.

**2.2.1 Forward method—Daily instantaneous.**   The cohort method (Section 2.2.1) can be formally interpreted as finding an adjusted deaths figures recording the number of deaths associated with cases today, which we will denote $\tilde{d}(t)$, and dividing this by the number of cases. That is,

$$CFR_{\text{cohort}}(t) = \frac{\tilde{d}(t)}{C(t)}, \tag{3}$$

where $C(t)$ is the number of cases on day $t$. Here, $\tilde{d}(t)$ is calculated by counting the number of deaths associated with COVID-19 among individuals testing positive on day $t$. This can be considered as summing the number of deaths on each day where the individual tested positive

on day $t$, across all future days, i.e.

$$\tilde{d}(t) = \sum_{x=t}^{\infty} (\text{number of patients dying on day } x \text{ who first tested positive on day } t). \quad (4)$$

To develop a probabilistic forward method, we need to approximate the summand on the right hand side of Eq (4) using time series data. For brevity, we will use the notation

$$\bar{d}_t(x) = (\text{number of patients dying on day } x \text{ who first tested positive on day } t), \quad (5)$$

such that $\tilde{d}(t) = \sum_{x=t}^{\infty} \bar{d}_t(x)$. Writing $\bar{d}_t(x)$ in terms of probabilities, we have

$$\bar{d}_t(x) = d(x)P(I = t | D = x). \quad (6)$$

where $d(x)$ is the observed number of deaths on day $x$, $I$ is the day of the first positive test, and $D$ is the day of death. This can be rewritten as

$$\bar{d}_t(x) = d(x)\frac{P(I = t \cap D = x)}{P(D = x)} \quad = d(x)\frac{P(D = x | I = t)P(I = t)}{P(D = x)}$$

$$= d(x)\frac{P(D = x | I = t)P(I = t)}{\sum_{y=0}^{x} P(D = x | I = y)P(I = y)}. \quad (7)$$

The probability of dying on day $x$ given infection on day $y$ is the probability that the delay distribution from testing to death is between $x - y$ and $x - y + 1$, which we denote $g(x - y|y) = F_\theta(x - y + 1|y) - F_\theta(x - y|y)$ (where $F_\theta(z|y)$ is the cumulative distribution function of the testing to death delay, given that testing happens on day $y$), multiplied by the probability of dying giving infection on day $y$, $CFR_f(y)$. Since we have the full historic testing data, the probability of testing positive on day $y$ is given by the number of first positive tests on day $y$ divided by the total number of first positive tests, i.e.

$$P(I = y) = \frac{C(y)}{\sum_{z=0}^{\infty} C(z)}. \quad (8)$$

Therefore, we have

$$\bar{d}_t(x) = d(x)\frac{g(x - t|t)CFR_f(t)\frac{C(t)}{\sum_{z=0}^{\infty} C(z)}}{\sum_{y=0}^{x} g(x - y|y)CFR_f(y)\frac{C(y)}{\sum_{z=0}^{\infty} C(z)}} = d(x)\frac{g(x - t|t)CFR_f(t)C(t)}{\sum_{y=0}^{x} g(x - y|y)CFR_f(y)C(y)}, \quad (9)$$

which leads to

$$CFR_f(t) = \frac{\sum_{x=t}^{\infty} d(x)\frac{g(x - t|t)CFR_f(t)C(t)}{\sum_{y=0}^{x} g(x - y|y)CFR_f(y)C(y)}}{C(t)} = \sum_{x=t}^{\infty}\frac{d(x)g(x - t|t)CFR_f(t)}{\sum_{y=0}^{x} g(x - y|y)CFR_f(y)C(y)}. \quad (10)$$

To solve this exactly, we would have to construct an iterative model for $CFR_f$ (which would lead to an impractical increase in computational cost relative to the backward method). However, although we are interested in the daily CFR, we would not expect the CFR to vary substantially on a daily basis (aside from weekend effects in the testing rates and stochastic noise). Therefore, to simplify the model we can assume that on the right hand side of Eq (10) the values of $CFR_f(x)$ are constant for all $x$. Therefore, we can approximate this by

$$CFR_f(t) \approx \sum_{x=t}^{\infty} \frac{d(x)g(x-t|t)}{\sum_{y=0}^{x} g(x-y|y)C(y)}, \tag{11}$$

which has comparable computational cost to the backward method. By making this assumption on the right-hand side, we effectively smooth through the variation in the daily CFR. In reality, when using the cohort method, we observe substantial daily variation in the CFR. However, this is down to noise in the data and weekend effects, whereby fewer tests are performed at weekends, perhaps biasing the sample to more severe cases. By smoothing through this short-term variability, this approximation gives a reasonable indicator of the general trend.

This formula for $CFR_f(t)$ allows us to calculate the case fatality risk using time series data. This resulting method is similar to the approach developed in [18], though the assumptions applied are different. The main difference between these models is the forward approach focuses on the case fatality risk among confirmed cases, whereas the Reich model attempts to estimate the infection fatality ratio.

**2.2.2 Forward method—Overall.** To estimate the overall CFR ($oCFR$) on a given day, we are interested in what proportion of total cases up to that day result in death. Taking a forward approach, we are therefore interested in what proportion of cases reported up to and including, the given day have died or will go on to die. That is

$$oCFR_f(t) = \frac{\text{total deaths among individuals testing positive before or on day } t}{\text{total individuals testing positive before or on day } t}. \tag{12}$$

The denominator here is easy to calculate, and is given by

$$\sum_{x=0}^{t} C(x), \tag{13}$$

where $C(x)$ is the number of cases on day $x$. For the numerator, we need to look at the number of deaths on every day between zero and infinity, and calculate what proportion of these deaths can be attributed to individuals testing positive before day $t$. That is, we can write the numerator as

$$\sum_{y=0}^{\infty} d(y)P(I \leq t|D=y). \tag{14}$$

Following the logic described in Section 2.2.1, this can be approximated as

$$\sum_{y=0}^{\infty} d(y)P(I \leq t|D=y) \approx \sum_{y=0}^{\infty} \frac{\sum_{z=0}^{t} d(y)g(y-z|z)C(z)}{\sum_{x=0}^{y} g(y-x|x)C(x)}. \tag{15}$$

Therefore we obtain

$$oCFR_f(t) \approx \frac{\sum_{y=0}^{\infty} \frac{\sum_{z=0}^{t} d(y)g(y-z|z)C(z)}{\sum_{x=0}^{y} g(y-x|x)C(x)}}{\sum_{w=0}^{t} C(w)}. \tag{16}$$

**2.2.3 Forward method—Weekly instantaneous.** The daily instantaneous CFR is a useful measure of the temporal trends in the CFR. However, there can be a lot of noise in the daily estimates, driven by stochasticity in the underlying process and day-of-week biases in the data streams (for example, testing capacity reduced at weekends or fewer deaths reported on weekends). This issue is particularly pertinent when the number of cases is low. One way to account for this, whilst still capturing the instantaneous CFR, is to look at weekly rather than daily estimates. The aim of a weekly CFR estimate is to capture the case fatality risk of either individuals testing positive during that week or dying during that week (depending on whether a forward or backward approach is used). Using a weekly CFR should remove the day-of-week effects, so that any temporal changes are genuine trends.

Under a forward approach, the weekly CFR is the number of individuals that test positive in a given week that go on to die, divided by the number of individuals testing positive during this week, i.e.

$$wCFR_f(\tau) = \frac{\tilde{D}(\tau)}{C(\tau)}. \tag{17}$$

The number of individuals testing positive in week $\tau$, $C(\tau)$ is given by $C(\tau) = \sum_{x=0}^{6} c(7\tau + x)$. Here, $\tau$ is a weekly index, starting from the first day considered in the time series. Multiplying $\tau$ by 7 gives the first day of each week. The number of individuals testing positive in week $\tau$ that go on to die, $\tilde{D}(\tau)$, is given by

$$\tilde{D}(\tau) = \sum_{x=0}^{6} \tilde{d}(7\tau + x) = \sum_{x=0}^{6} \sum_{y=7\tau+x}^{\infty} \frac{d(y)g(y - 7\tau - x|7\tau + x)CFR_f(7\tau + x)c(7\tau + x)}{\sum_{z=0}^{y} g(y - z|z)CFR_f(z)c(z)}. \tag{18}$$

Making the assumption of small variation in CFR over small timescales, this gives the weekly forward CFR as

$$wCFR_f(\tau) = \frac{\sum_{x=0}^{6} \sum_{y=7\tau+x}^{\infty} \frac{d(y)g(y - 7\tau - x|7\tau + x)c(7\tau + x)}{\sum_{z=0}^{y} g(y - z|z)c(z)}}{\sum_{x=0}^{6} c(7\tau + x)}. \tag{19}$$

**2.2.4 Backward method—Weekly instantaneous.** We can also estimate the weekly CFR using a backward approach. Through this, we obtain

$$wCFR_b(\tau) = \frac{D(\tau)}{\tilde{C}(\tau)}, \tag{20}$$

where $D(\tau)$ is the number of deaths in week $\tau$ and $\tilde{C}(\tau)$ is the number of cases that could have died in week $\tau$. $D(\tau)$ is simple to extract from the data, and is given by $D(\tau) = \sum_{x=0}^{6} d(7\tau + x)$. $\tilde{C}(\tau)$ is given by

$$\tilde{C}(\tau) = \sum_{x=0}^{6} \tilde{c}(7\tau + x) = \sum_{x=0}^{6} \sum_{y=0}^{7\tau-x} c(y)g(7\tau + x - y|y). \tag{21}$$

Therefore, the weekly backward CFR is given by

$$wCFR_b(\tau) = \frac{\sum_{x=0}^{6} d(7\tau + x)}{\sum_{x=0}^{6} \sum_{y=0}^{7\tau+x} c(y)g(7\tau + x - y|y)}. \tag{22}$$

## 2.3 Relating the forward and backward methods

As two measurements of the CFR, there should be some agreement between the forward and backward methods. We might not get perfect alignment of the daily CFR, since one looks at the given day as the day of testing and the other considers the given day as the day of death, but the general trends should be consistent. In this section, we consider the special case where the delay from testing to death is constant rather than drawn from a random distribution. In such a case, we find that the backward method can be transformed to be identical to the forward method, for the daily-instantaneous CFR. For simplicity, here we assume that the delay from testing to death is independent of testing time.

Consider a constant delay distribution such that

$$g(x) = F_\theta(x+1) - F_\theta(x) = \begin{cases} 1, & \text{if } x = \lambda \\ 0, & \text{otherwise.} \end{cases} \tag{23}$$

Under such a distribution, the forward CFR reduces to

$$CFR_f(t) = \frac{d(t+\lambda)}{C(t)}, \tag{24}$$

and the backward CFR reduces to

$$CFR_b(t) = \frac{d(t)}{C(t-\lambda)}. \tag{25}$$

On the given day, the two CFR methods will give different estimates. However, by finding the backward CFR on day $t + \lambda$, we have

$$CFR_b(t+\lambda) = \frac{d(t+\lambda)}{C(t)}, \tag{26}$$

which is the same as $CFR_f(t)$. Therefore, under a constant death delay, the backward CFR can be transformed into the forward CFR by lagging the data by the length of the constant delay.

For general distributions, we cannot directly transform the backward CFR into the forward CFR through a simple lag. However, we investigate shifting the general backward CFR by the expected duration of the death delay, to see how this compares with the forward CFR.

## 2.4 Uncertainty quantification

When interpreting the CFR, we can consider the observed number of deaths as a random sample from a binomial distribution, with number of trials equal to the number of cases on that day, following [11, 23]. Classing deaths as the event of interest, the case fatality risk is the probability of the event. Finding the probability that maximises the likelihood of the observed number of deaths allows us to find the mean estimate for the case fatality risk on each day. By constructing a likelihood function for the data, we can now quantify uncertainty around this point estimate, which illustrates potential other values for the daily CFR that would not be inconsistent with the data. That is, the mean estimate is the CFR which best represents the data on that day, but since we only have one observation each day, the actual fatality risk could have been different, with the observation affected by random noise. To quantify this uncertainty, we will use the likelihood ratio method (Wilks test). By using the chi-squared cutoffs at 95%, we can find the 2.5% and 97.5% percentiles for the daily CFR. The benefit of this method is that uncertainty grows when fewer cases are observed. That is, CFR estimates when prevalence is lower will be far more uncertain than during high prevalence phases. This is because the likelihood function becomes flatter when the number of trials is smaller. This is beneficial, as when the number of cases is higher, random noise should play a smaller part in the number of observed deaths.

Therefore, we wish to make the following assumptions. For the adjusted deaths value $\tilde{d}(t)$, i.e. the deaths corresponding to tests on day $t$, we can assume this (subject to rounding to the nearest integer) is drawn from a binomial distribution with $C(t)$ trials and probability $CFR_f(t)$:

$$\left\lceil \frac{\lfloor 2\tilde{d}(t)\rfloor}{2}\right\rceil \sim \text{Bin}\Big(C(t), CFR_f(t)\Big). \tag{27}$$

For the number of observed deaths on each day $d(t)$, we can assume this is drawn from a binomial distribution with $\tilde{C}(t)$ trials (subject to rounding to the nearest integer) and probability $CFR_b(t)$:

$$d(t) \sim \text{Bin}\left(\left\lceil \frac{\lfloor 2\tilde{C}(t)\rfloor}{2}\right\rceil, CFR_b(t)\right). \tag{28}$$

since $d(t)$, $\tilde{d}(t)$, $C(t)$, and $\tilde{C}(t)$ are observed, we can use these to construct likelihood functions for observing these data given $CFR_f(t)$ and $CFR_b(t)$. From these, we can estimate the maximum likelihood estimator for these case fatality risk and uncertainty, as described above.

This method provides a good indicator of the uncertainty in a given estimate of the CFR. However, this does not consider correlation between the CFR on nearby days, so perhaps overestimates the uncertainty when investigating temporal trends. When applied to overall and weekly CFR estimates, this should provide a more robust measure of uncertainty, since correlation between weeks will be reduced. Alternative methods for estimating uncertainty may be better at capturing this correlation, but we do not consider these here.

## 3 Results

### 3.1 Method comparison

In this section, we compare estimates from the methods presented above for generating the case fatality risk across care home residents in England. Here, we only use the PHE 28-day deaths data as an example. CQC deaths data are presented in Section 3.2.

Note that, the delay from testing to death is likely to vary over time. To capture this variation, we estimate the delay on a monthly basis (see S2 Text), so that the delay depends on the

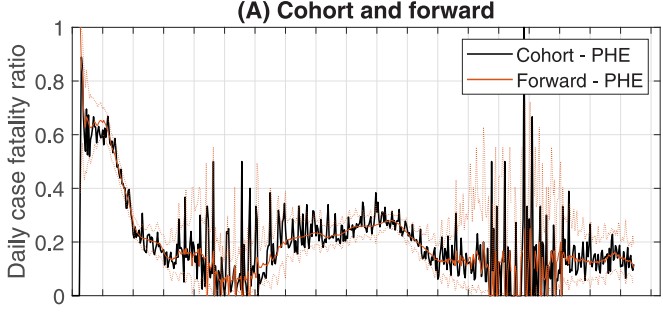
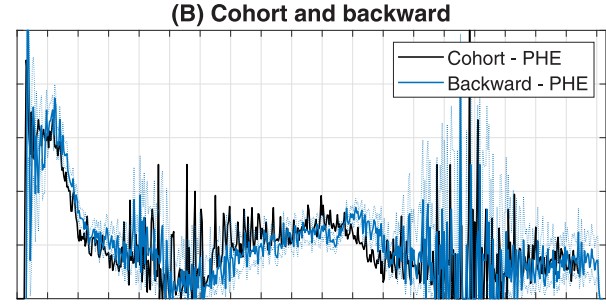

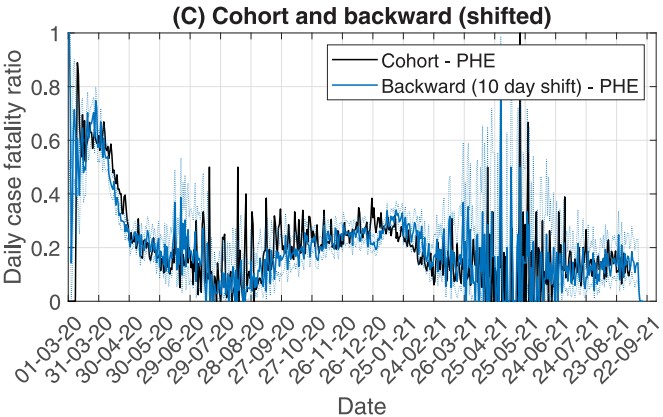
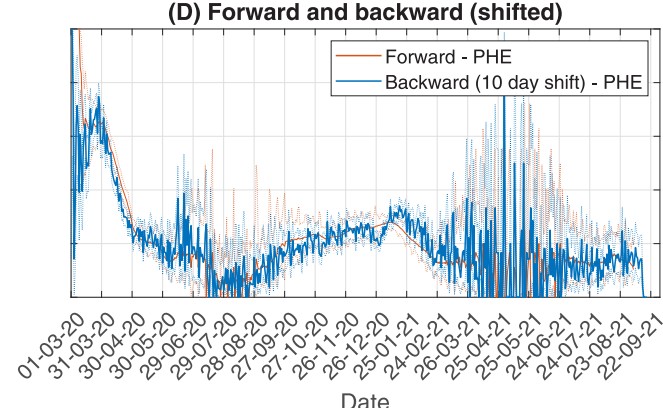

**Fig 2. Comparing the methods.** Top left: cohort versus forward. Top right: cohort versus backward. Bottom left: Cohort versus shifted backward. Bottom right: Forward versus shifted backward. Dashed lines indicate the 95% confidence intervals of the daily CFR estimates. In the shifted backward results, we shift the daily estimates to 10 days prior, reflecting the average delay from testing positive to death.

month in which an individual tests positive. Using this temporally changing delay distribution allows us to estimate the CFR more accurately.

**3.1.1 Forward versus backward versus cohort.** Here we investigate the three different approaches for estimating the CFR. The cohort method acts as a "ground truth" of the actual CFR observed on each day. Through comparing this to the forward and backward methods we can evaluate their performance at estimating the CFR.

Fig 2 shows the daily estimates for each of the methods. Cohort indicates the cohort method applied to the individual-level line-list data. Forward is the forward approach applied to the time series data. Backward is the backward approach applied to the time series data. Backward —shifted is the solution to the backward approach, shifted back 10 days, in line with the expected value of the death delay distribution. As we consider only deaths which occur within 28 days of the first positive test result, the forward approach cannot reliably estimate the CFR for the last 28 days of data, so these dates have been removed. The backward approach however can still estimate over these days, though the last four days are removed to handle reporting lags in deaths data.

The upper two graphs compare the forward and backward approaches, respectively, to the cohort results. Panel (A) compares the forward method to the cohort method. We see good agreement between the two methods. However, as expected, the forward method leads to a much smoother temporal trend, since this smooths through the day-to-day variation. This smoothing can make it easier to identify the temporal trend. Panel (B) then compares the

backward method to the cohort. The temporal trends appear very similar, in both shape and daily variation. However, the backward method appears shifted in time to later dates. This is because the backward method looks at CFR by date of death, as described in Section 2.3.

To adjust for this asynchrony in the backward method, we can shift the data to earlier dates by the mean delay from testing to death, which is 10 days (averaged across the whole pandemic). To do this, we shift the backward solutions to the left by 10 days. The results of this are compared to the cohort method in panel (C). This leads to substantially improved agreement between the cohort and backward method. Therefore, by shifting the backward method, we can interpret the output as an approximation to the cohort method. However, the shift method would not work as well if the delay distribution from testing to death changes substantially across the pandemic, such as if testing policy changed or a new variant led to faster mortality, since we are shifting all estimates by a single value. In panel (D), we compare the cohort method to the shifted backward method. The two methods have good agreement in both temporal trend and uncertainty, though the forward method has reduced day-to-day variation.

To further assess the accuracy of the backward and forward approximation methods, relative to the cohort method, we calculate the relative error (S4 Text). The forward method provides the best approximation to the cohort method, with mean relative error equal to -0.0132. The shifted backward comes second, with mean relative error equal to -0.0487. The backward method has the worst performance, with mean relative error equal to -0.0919.

**3.1.2 Daily versus weekly.** For the instantaneous CFR we have developed methods for estimating either daily or weekly values. The weekly method can also be used to aggregate at other temporal resolutions, such as monthly. In Fig 3 we compare the estimates at both daily

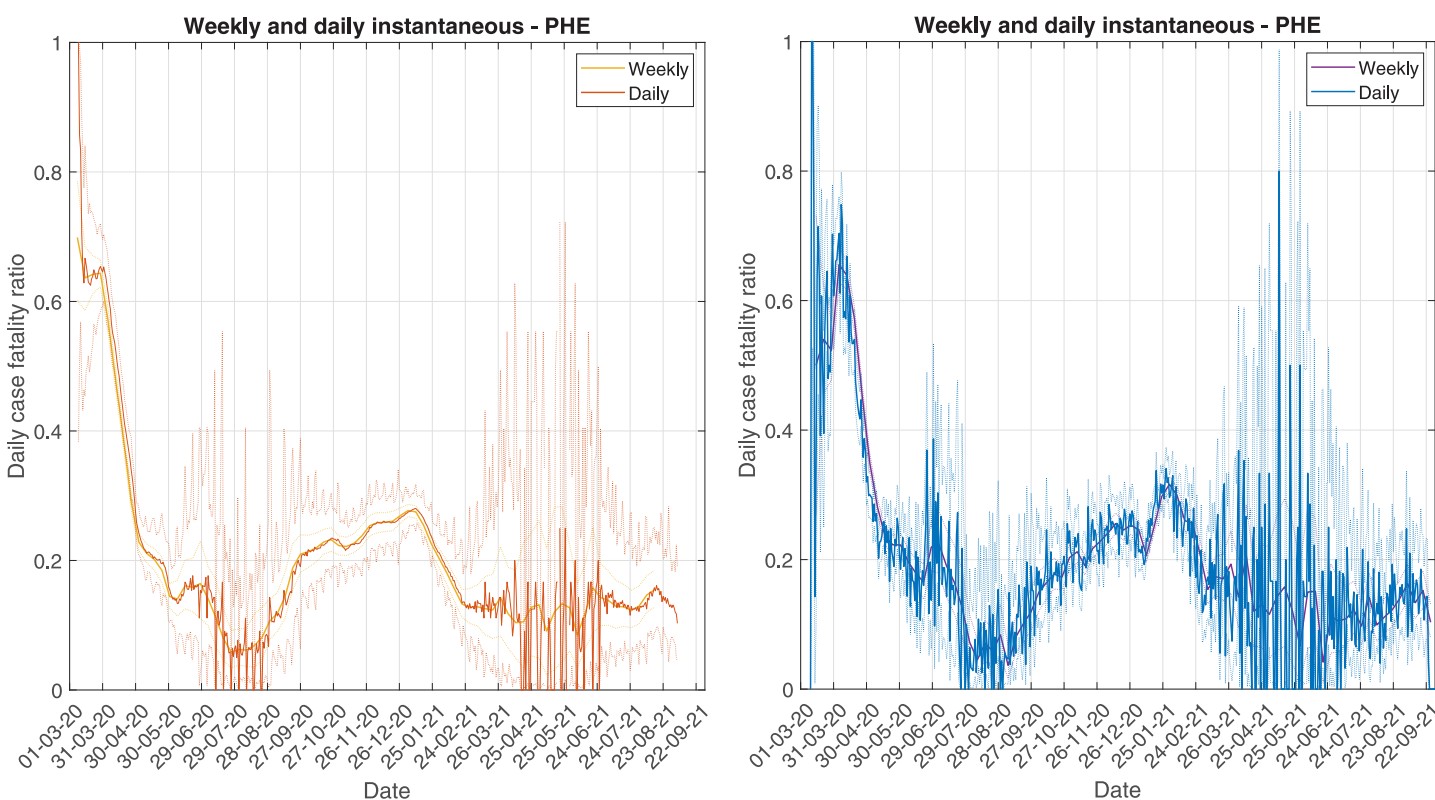

**Fig 3. Comparing weekly instantaneous CFR to daily instantaneous CFR, using PHE data.** Left panel shows the forward methods and right panel shows the backward methods.

and weekly resolutions with the forward and backward methods. The values are slightly out of sync since the weekly value reflects the CFR across that week rather than on any given day during that week. However, the temporal trends are largely consistent at each resolution. The weekly method is less noisy and has tighter uncertainty, since the sample sizes each week are larger. Therefore, the temporal trend in the weekly method can be easier to interpret than the daily method, particularly when case numbers are low, such as after 01/03/2021. However, the uncertainty in the weekly CFR does not reflect the volatility of the daily values (due to the smaller numbers of daily observations).

**3.1.3 Instantaneous versus overall.** The CFR can either be considered as an instantaneous estimate or overall estimate. The overall estimate does not necessarily give a good indicator of the current risk, since it is heavily biased by historic values. In Fig 4, we plot the overall and the daily instantaneous estimates, using the forward and backward methods. During the first wave, we see that the lack of testing led to very high values in both estimates. This prolonged period where deaths made up a very high proportion of tests leads to the overall CFR overestimating the current risk, since this causes a significant upwards bias. To account for this, we could remove the first wave from the modelling. However, we would then encounter issues on how to truncate the data streams, since there is no prolonged phase with no cases and no deaths. Looking at the temporal trend in the overall CFR, we see a declining risk since the first wave. However, this again is driven by the high CFR during the first wave. As the daily analysis has shown, the CFR dropped over summer, when prevalence was very low, before steadily rising into the winter. In 2021, the CFR then started to drop, since the vaccination

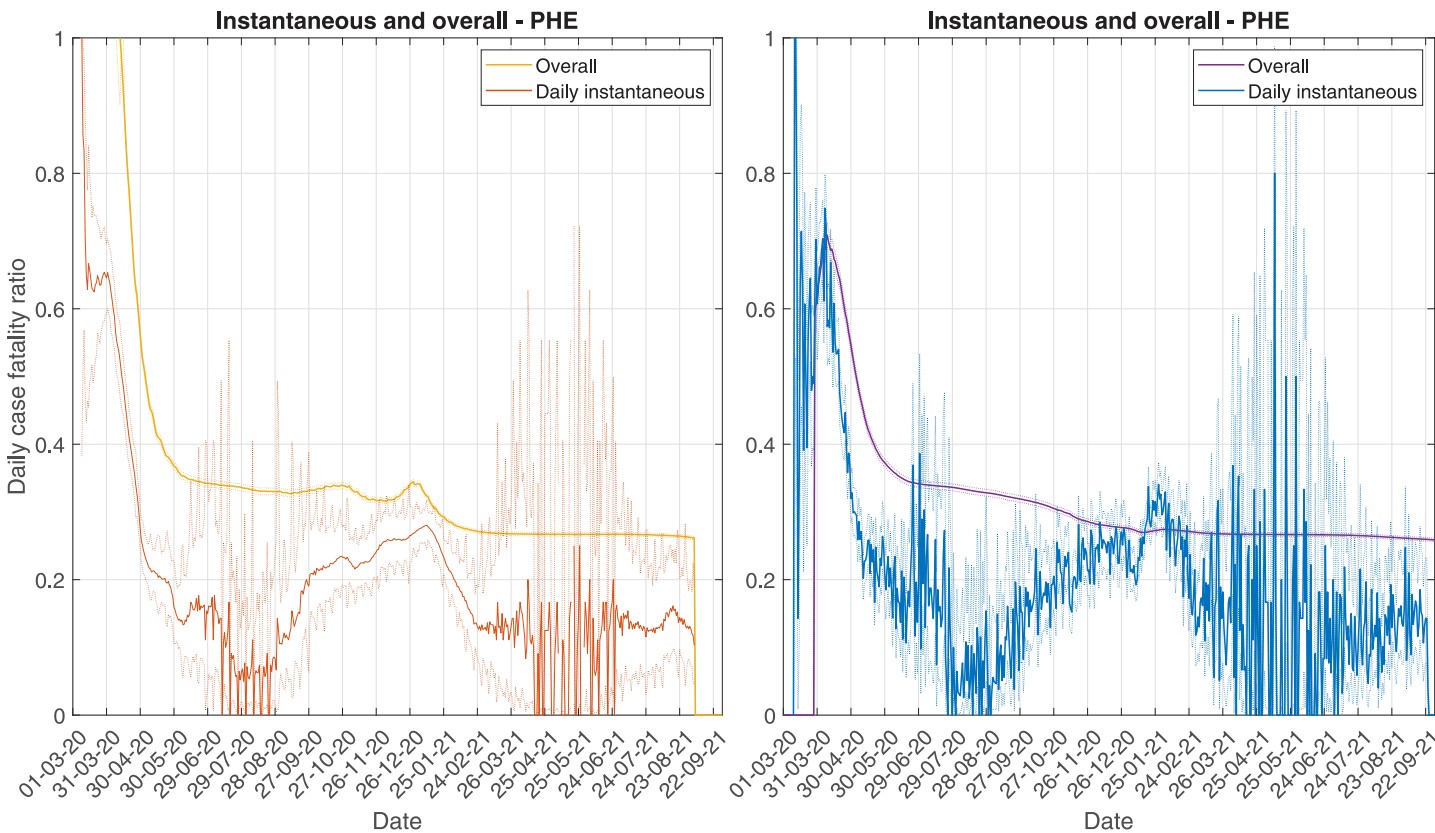

**Fig 4. Comparing daily instantaneous CFR to daily overall CFR, using PHE data.** The left panel shows the forward methods, and the right panel shows the backward methods.

programme was rolled out to care home residents and the prevalence started dropping due to lockdown. The overall estimate does not shine a light on any of these trends, instead suggesting the false signal that the risk has been steadily declining. One advantage however of using the overall method is the uncertainty is significantly reduced. Since this is looking at the CFR of all cases up to a given date, the sample size is very large, so we can be confident that the estimate is close to the true overall CFR. The main issue is in how this estimate is interpreted—it is a calculation of the mortality of all historic cases, rather than a reflection of current mortality risk. Although a measure of overall mortality, the overall CFR may be unreliable for comparing overall mortality across sources that follow distinct epidemic trends, since the estimates may be subject to difference biases (see Section 3.2.5).

## 3.2 Operational results

In Section 3.1 we compared the different methods for estimating the CFR. In this section, we apply these methods for different operational purposes, looking at how different data sources affect the CFR estimates, how the CFR varies across age groups and regions, and how the type of care provided affects the CFR.

**3.2.1 Temporal trends in the care home CFR.**   In Fig 2, we plotted the temporal trend in the daily instantaneous CFR across the whole pandemic. Combined with Fig 3, which contains the weekly CFR estimates across the pandemic, we can investigate the temporal trends in severity. Early in the pandemic, the CFR estimates are very high (over 60%). This is likely driven by lack of testing, leading to unreliable estimates. From April, more routine testing was started, which reduced the CFR, dropping to around 15% by May 2020. Over summer 2020, the CFR dropped further to below 10%, likely driven by a combination of epidemic phase bias [24], false positives [25, 26], and high-frailty individuals potentially dying earlier in the pandemic.

The CFR jumped up in September 2020 to around 20%, before rising steadily to 30% by December 2020. The main driver of this steady rise seems to be different timings in outbreaks across the English subregions (Section 3.2.5).

In late December, the CFR increased again, which was likely driven by the increased severity of the alpha variant relative to the previously circulating variants [19], though could also be driven by the increased growth rate of the alpha variant [27], which will lead to epidemic phase bias [24].

From January 2021, the CFR rapidly declined to 13% by the end of February 2021. The major driver here is vaccine effect, which reduces the severity of confirmed cases. However, epidemic phase bias and false positives could be contributing to the decline, so the precise influence of vaccination on CFR cannot be readily extracted.

Since March, the CFR has plateaued at 13%. The uncertainty in the estimate has increased/decreased in line with prevalence, but the central estimate has remained relatively constant. This suggests mortality risk remains relatively high amongst this cohort, though substantially better than pre-vaccine.

During August 2021, there appears to have been a slight bump in the CFR, where it increased slightly before decreasing back down to the 13% plateau. This occurs around the same time as a rise and fall in cases in care home residents. Therefore, this bump is likely caused by the epidemic phase bias [24].

**3.2.2 Data source variation.**   When estimating the CFR, we have two unique data sources for deaths: PHE time series and CQC time series. Here, we compare the temporal trends in the CFR in English care homes across the two data sets (Fig 5).

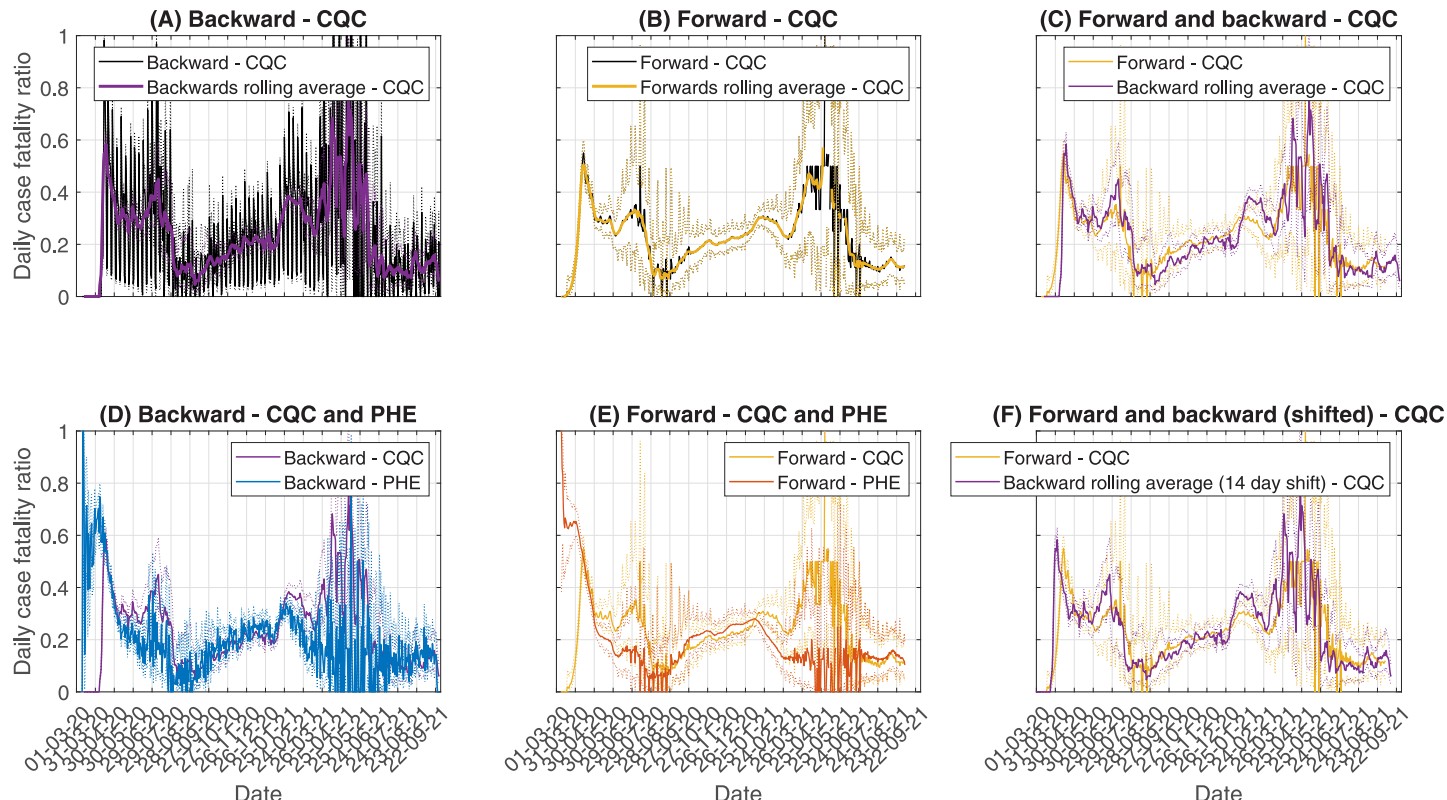

**Fig 5. Comparing estimates derived from the CQC data to the PHE data.** Since the CQC data uses date of report, we have strong weekend effects in the backward method, so also look at the 7-day rolling average (A). The forward method does not exhibit the weekend effects, since this method smooths through nearby CFR estimates, so the 7-day rolling average is not necessary here (B). When comparing forward and backward (C) and (F), we use a 14-day time shift, since this is the expected delay used from testing positive to reporting death. In (D) and (E) we plot the PHE estimates for comparison.

What immediately becomes apparent when using the CQC data is strong daily fluctuations in the backward CFR, with estimates fluctuating between 0.05 and 0.4 in the space of a couple of days. This makes it hard to identify any temporal trends. Instead, we can consider a smoothing of the temporal trends, such as calculating the rolling average. In Fig 5, we consider the 7-day rolling average, since this allows any day-of-week effects to be smoothed. The resulting curve is much smoother and allows us to investigate the temporal trends. Comparing this to the PHE backward method, we see very similar temporal trends in the CFR for the first year of the pandemic. There is a slight lag in the early stages of the pandemic since this is before CQC started reporting deaths via COVID-19 status. The CQC CFR experiences potentially slower growth during Autumn 2020, offset by faster growth in December 2020, though the confidence intervals of the two curves overlap throughout this period.

Using the forward method, we do not observe the day-of-week based fluctuations in the CFR. This is because of the assumption made in Section 2.2.1, where we assume that CFR variations over small timescales are negligible. Comparing the rolling average to the raw CFR, we therefore observe very little difference, since the raw CFR is smoothed during the calculation. Comparing the forward method to the PHE data, we again see similar trends for most of the time series.

Comparing the forward and backward methods for the CQC data, we again observe a shift in the time series. When approximating the delay distribution from testing to death report, we take the shape parameter from the testing to death delay, but increase the scale parameter to

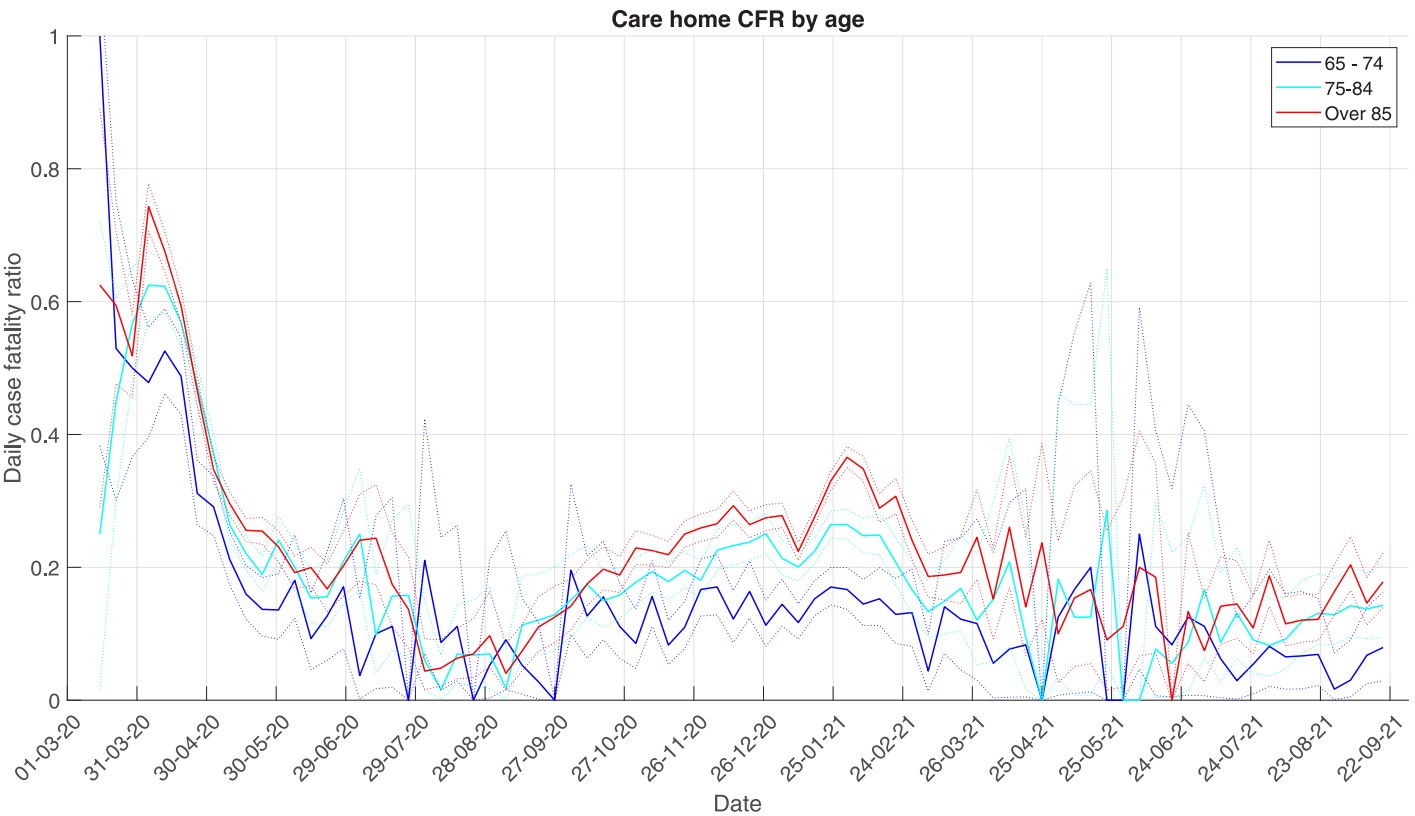

**Fig 6. Comparing the CFR within different age groups of care home residents, using the weekly backward method and PHE data.**

increase the expected delay by 4 days, to allow for delays in death reporting. Therefore, the expected delay is around 14 days. Shifting the backward method by 14 days earlier, we see good overlap between the forward and backward methods, again showing that transforming the backward methods by the expected delay allows us to compare the forward and backward method.

Whilst the CQC CFR has been consistent with PHE across the first year of the pandemic, in early 2021 the trends have significantly diverged as the prevalence dropped. After May 2021, as prevalence started to rise again, the trends have become consistent.

**3.2.3 Impact of age.** With the severity profile of COVID-19 rapidly increasing with age, looking at the CFR in different age groups of care home residents may be important. Here we stratify residents into three age groups: 65–74, 75–84, and over 85. In Fig 6 we plot the weekly instantaneous CFR, calculated using the backward methods and PHE deaths data. Here, we see a clear increase in CFR as the age group increases, with a December 2021 peak in the oldest age group around 35%, 25% in the middle age group, and 15% in the youngest age group. These differences between the CFR across age groups are significant at the 95% level, as shown by the risk ratio calculation in Fig A1 in S5 Text. All age groups see CFR declining from January 2021. Prior to this, the CFR in the 65–74 age group was constant. The CFR in the other two age groups increased gradually from September 2020.

**3.2.4 Type of care.** A potential covariate affecting the CFR is the type of care given within the care home. Under the classification of the CQC, we have both residential care homes without nursing and residential care homes with nursing. Where residents have regular nursing care, we would expect a demographic with higher frailty, due to the regular care needs.

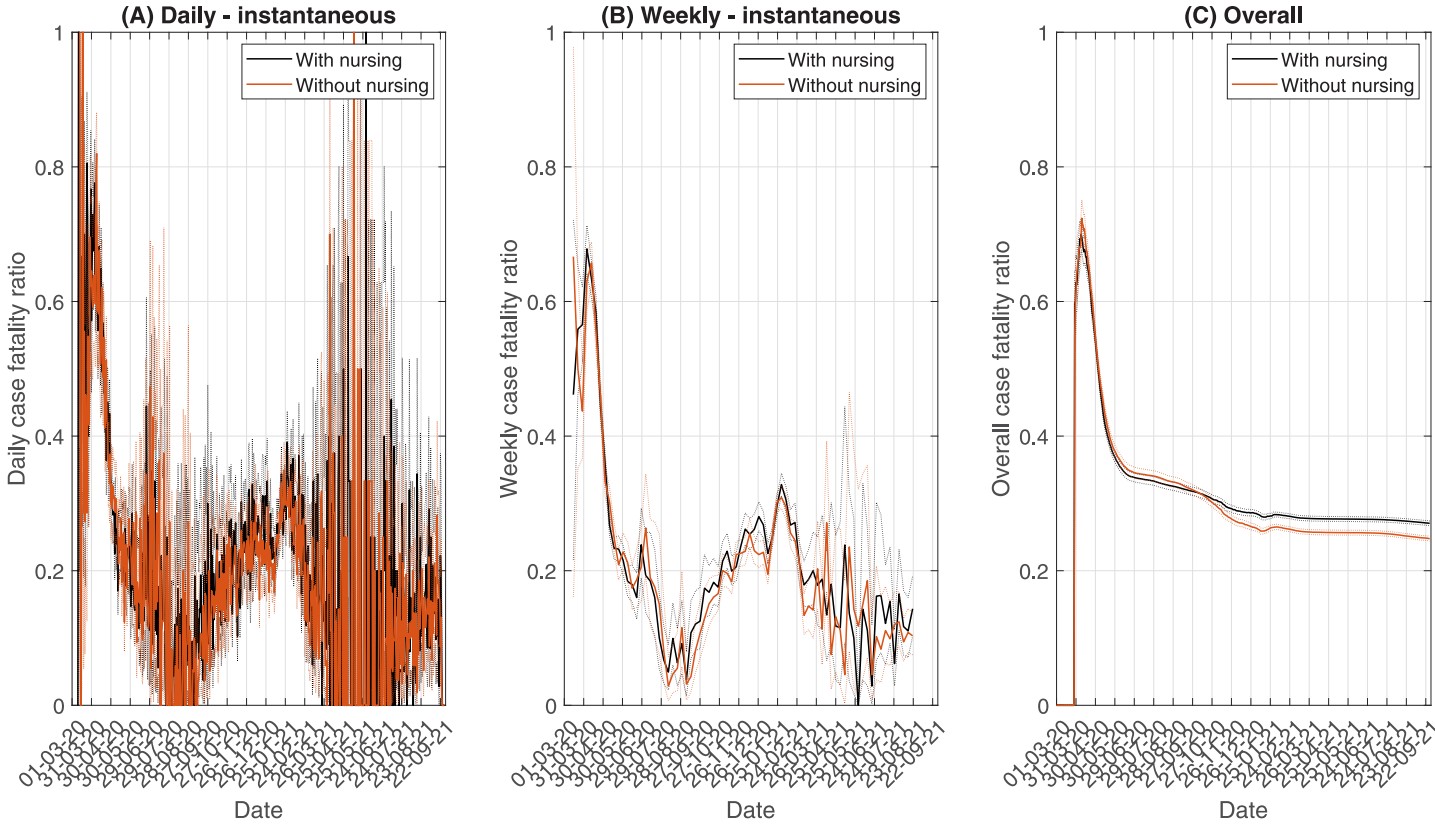

**Fig 7. CFR based on type of care: With nursing or without nursing, using PHE data and backward methods.** (A) shows the daily instantaneous CFR, (B) shows the weekly instantaneous CFR, and (C) shows the overall CFR over time. Dashed lines indicate the 95% confidence intervals.

Therefore, a higher CFR might be observed within such care homes. In this section, we investigate how type of care influences the CFR. Each test and death can be linked to a specific care home. Through this linkage, we identify type of care through a master list of care homes, which describes the location and type of care provided. This enables us to construct two independent time series, one for tests and deaths with nursing and the other for tests and deaths without nursing, from which we can estimate the CFR with and without nursing.

Looking at the daily instantaneous CFR (Fig 7A), the temporal trends are similar for both types of care, with no significant difference in the daily CFR. The central estimate of the CFR for care homes with nursing is consistently slightly higher than in homes without nursing, suggesting a potential increased risk. However, the high level of uncertainty makes this unclear. To reduce uncertainty, we can use the weekly instantaneous CFR (Fig 7B). Here, the central estimate is clearly higher for care homes with nursing than without nursing. However, the difference is only small, and is only significant (at the 95% level) for one week across the whole epidemic. Looking at the overall CFR (Fig 7C), we see that in the last few months, the overall CFR has become significantly higher in care homes with nursing than without, with a CFR around 28% relative to 25%. Therefore, although at a given point in time the risk of mortality in care homes with nursing is not significantly larger, over the time course of the epidemic there has been increased mortality in such care homes. However, this small difference in the CFR between the two types of care suggest that it does not have a substantial impact on residents' outcomes. These results are verified using risk ratio calculations in Fig A2 in S5 Text.

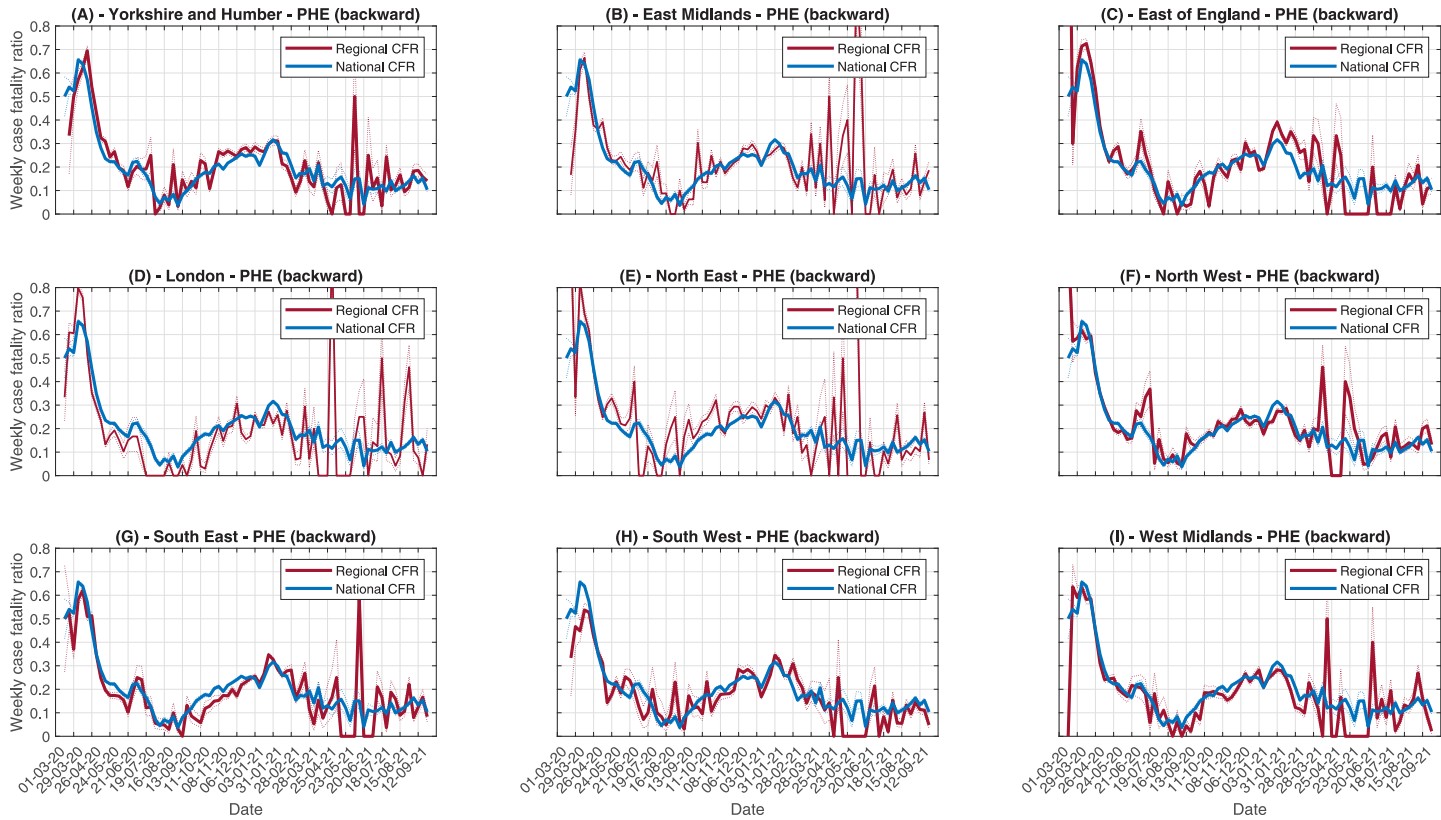

**Fig 8. Comparing weekly CFR trends across English regions, using PHE data and the backward method.** Each panel contains the regional trend and the national trend.

**3.2.5 Regional variation.** Between countries, we might expect substantial differences in care home case fatality risks, as different countries may have different care policies and different demographics requiring care. Similarly, within a country, different regions could have variations in their care home CFR. This could be driven by differences in care provisions, demographic differences between regions affecting the frailty and age distribution of residents, and the severity of local outbreaks. In this section, we investigate how the case fatality risk varies across different NHS regions in England. We analyse both temporal trends for each region, using instantaneous CFR estimates, as well as the overall CFR. Temporal trends allow us to investigate whether different regions have different trends, some of which is to be expected, since the CFR is linked to local incidence, and different regions have different epidemic time series. The overall trends allow us to compare overall mortality across the regions, but may not be robust since testing behaviour has changed substantially between the first and second waves, which can bias the overall CFR.

Looking at regional data, the number of observed cases and deaths each day is much lower than at national level, because of the smaller number of care home residents in each region. Therefore, we opt to use the weekly instantaneous CFR rather than daily, since this will provide a more stable signal with lower uncertainty. In Fig 8 we compare the regional weekly CFR, estimated using the PHE data with the backward method, to the national average, to identify where regional trends differ from national trends (Fig 9 shows the same for CQC deaths data). These figures are supported by risk ratio analysis in Figs A3 and A4 in S5 Text. For most regions, the regional CFR follows a similar trend to the national CFR, with high values during

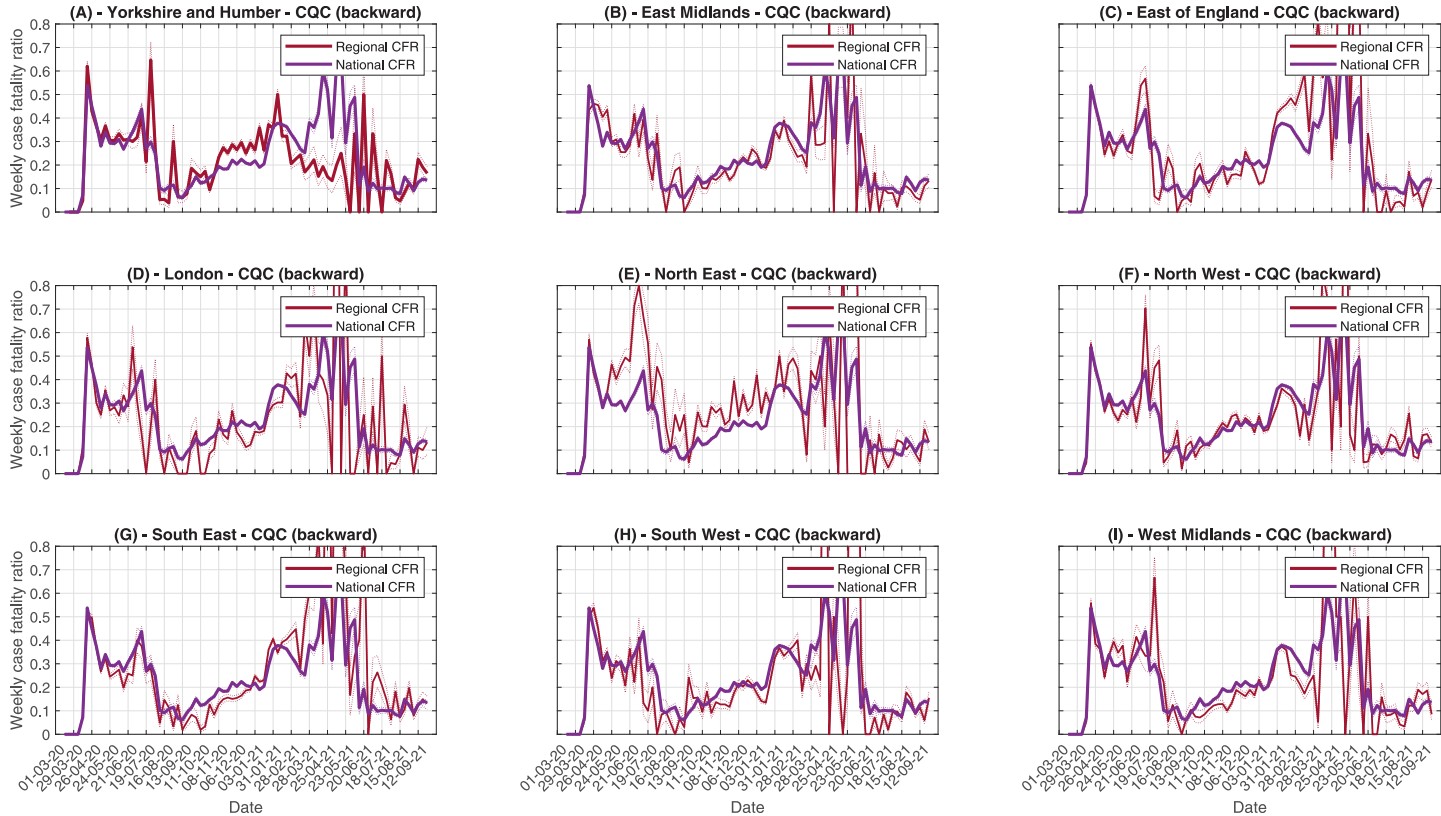

**Fig 9. Comparing weekly CFR trends across English regions, using CQC data and the backward method.** Each panel contains the regional trend and the national trend.

the first wave, a trough during summer as the testing volume increased (and some of the frailest individuals had already died during the first peak), rising during the second wave, and declining in 2021. In the North East, the CFR increased in Autumn 2020 much earlier than the national average, correlating with more cases in the local care homes. Over the most recent months, the local CFR agreed with the national CFR. However, the decline in the North East has not been as pronounced as the national CFR. On the other hand, the CFR in London has been substantially lower than the national CFR across most of the pandemic, which could be caused by London being hit particularly badly during the first wave, when testing rates were lower, with many high frailty residents potentially dying without becoming a confirmed case. The remaining regions follow the national trend more closely, with some discrepancies based on the timings of the local outbreaks. In many regions, the CFR from October to January is much flatter than the national CFR, suggesting there is little temporal change in CFR on these spatial scales. This suggests that the national trend, whereby CFR is gradually increasing, could be driven by the different timings in the regional outbreaks, so that the CFR is averaged across high CFR and low CFR regions, rather than any genuine increase in the mortality risk. Most regions observe a small rise in CFR in December, which could be driven by increased severity of the B.1.1.7 variant [19]. The CQC deaths data (Fig 9) lead to similar conclusions. Using risk ratio calculations (Fig A4 in S5 Text), all regions except North East and Yorkshire have periods of both significantly increased and decreased risk relative to the national average. From the trends in Fig 9, there is no consistent pattern. These significant differences are likely driven by slight differences in the reporting delay distribution for care homes in each region compared

**Table 1. Overall CFR as of 27/09/2021 in each region of England, using the backward method and PHE data.**

| Region | Overall CFR |
|---|---|
| Yorkshire and Humber | 0.258 (0.250, 0.265) |
| East Midlands | 0.248 (0.240, 0.256) |
| East of England | 0.282 (0.275, 0.289) |
| London | 0.266 (0.256, 0.276) |
| North East | 0.275 (0.265, 0.285) |
| North West | 0.247 (0.241, 0.253) |
| South East | 0.257 (0.251, 0.263) |
| South West | 0.241 (0.233, 0.249) |
| West Midlands | 0.247 (0.240, 0.255) |

to the national average, leading to the adjusted denominators being slightly mismatched. North East and Yorkshire however shows a consistently significantly increased risk relative to the national average during Summer 2020 and Spring 2021, which agrees with the analysis based on PHE data.

In Table 1, we report the overall CFR as of 27/09/2021 for each region in England. This provides a picture of the overall mortality with COVID-19 in care homes for each region. However, this does not provide a reliable comparison between regions. The instantaneous estimates in Figs 8 and 9 show that London has perhaps the lowest CFR across most of the time series. However, the overall CFR for London is the third highest. This seems to be driven by the overall CFR being more heavily biased towards the first wave in London, resulting in a higher overall CFR due to the lower rate of testing during the first wave, despite recent instantaneous estimates being far lower than many other regions. This demonstrates why instantaneous CFR estimates are more reliable than overall estimates.

## 4 Discussion

The case fatality risk is a key indicator for monitoring the impact of a pandemic. The high mortality of elderly individuals has posed a particular challenge for care homes in England. To help address this challenge, we have monitored the case fatality risk in English care homes throughout the pandemic. In this paper, we presented a variety of methods, both existing and novel, for estimating the case fatality risk. We discussed and compared these methods, before applying them to investigate the case fatality risk across English care homes. This work has been important, as prior to its implementation, the working assumption of care home CFR in England was taken to be as low as 10%, since it was estimated during the summer. However, our research suggested that at its peak the CFR reached 30%, which is a substantial increase, and shows the importance of investigating temporal trends in mortality risk.

The typical method for estimating the CFR is the "cohort" method, which requires line-list data linking cases to deaths. This method is easy to apply and interpret, but such data are often not readily available. When only time series data are available, the typical method is the "backward" method. Whilst this method is easy to apply to time series data, it is not as easy to interpret as the "cohort" method. To address this, we developed a novel "forward" method, which acts as an approximation to the "cohort" method that can be estimated from time series data rather than line-list data. Through comparing these methods, we found the forward method to be a good approximation to the cohort method. However, one of the assumptions in the construction of the forward method yields a smoother temporal trend than the cohort method, so the daily variability in CFR is not reflected, which is important when reporting varies between

weekday and weekend. However, the smoothing can make it easier to identify the overall trend. The backward method on the other hand captures a similar scale of daily variability, but appears to be temporally shifted forward in time. We have shown that this temporal shift is approximately equal to the mean delay from testing positive to death. Therefore, by shifting the backward method backward in time by this mean delay, we can approximate the cohort method.

These methods concern the daily-instantaneous CFR, which estimates the CFR on a given day (either by cases testing positive on that day or individuals dying on that day). This gives a high-resolution temporal trend, which would allow changes in CFR over small time scales to be picked up. However, if the number of observations is very low, these estimates can be highly uncertain with lots of day-to-day volatility, which can make it hard to interpret the temporal trends. To address this, we developed novel methods for estimating the instantaneous CFR with a weekly rather than daily aggregation, using both backward and forward methods. The resulting methods significantly reduce the uncertainty and volatility in the CFR estimates, making it easier to interpret the temporal trends. This can be vital during a pandemic, when we need to quickly identify whether, for example, a new variant is likely to lead to increased mortality in care home residents. In addition to weekly aggregating, these methods can be applied to any number of days, so fortnightly or monthly CFR estimates can also be generated, for example.

The final approach often used when investigating the CFR is the overall CFR. Where the instantaneous methods estimate the risk in a given time frame, the overall methods estimate the CFR up to a given point in time. Backward approaches are often applied to estimate the overall CFR. We therefore develop a novel forward approach for comparison. One advantage of the overall method is that through aggregating all mortality up to a given point in time, the number of observations is maximised, which reduces the uncertainty of estimates. However, since this considers all mortality to date, it is highly sensitive to changing testing rates over time. We have shown that in England, the initial instantaneous CFR was very high due to low testing rates. These very high initial values skew the overall CFR for all future time points, resulting in overall CFR estimates that do not provide much insight into the current mortality risk.

These conclusions regarding the relative effectiveness of each level of aggregation are context specific. For English care homes, the weekly resolution was sufficient to give easily interpreted trends whilst remaining flexible enough to detect temporal changes. In scenarios with higher daily incidence, the clarity of the daily resolution would improve, and therefore may be optimal for detecting temporal changes. Conversely, with lower incidence, even weekly resolution may be too fine, requiring monthly aggregation, for example. Instead of treating each estimate as independent, smoothing methods or multi-level approaches could be used to obtain smooth trends in the presence of high volatility. However, since this paper focused on methods for correctly adjusting the numerator and denominator in the CFR calculation, we did not consider these here.

To apply the forward and backward methods, we need to estimate the delay distribution from testing positive to death. To do this, we apply a truncation-corrected maximum likelihood estimation model to PHE line-list data (S2 Text). Through this, we have generated estimates for the mean delay (and standard deviation) from testing to death for individuals testing positive during each month of the pandemic. We observed that during the first wave the delay was quite short, since individuals were only tested if they were severely ill, due to shortage of tests. Since mass testing has been rolled out in care homes, the mean delay has increased (from around 8 days to 12 days). After this, the delay has remained relatively stable over time,

suggesting that the disease progression, conditional on severity, has not changed significantly with the emergence of new variants.

In application to COVID-19 mortality in care homes, we have shown that the presented methods can be used in conjunction with each other, to validate results and ensure overall trends are reliable. We see that the cohort, backward, and forward methods produce broadly similar results to one another, subject to smoothing and shifting. Additionally, different methods can be selected to address different objectives. To illustrate this, we applied the methods to investigate different aspects of the case fatality risk in care homes.

Firstly, we analysed two separate data sources for deaths data: PHE, by date of death, and CQC, by date of report. With the PHE data, it is possible to use the cohort method since the data are linked. For the CQC data however, only time series are available, illustrating why the backward and forward methods are powerful. We found that the two data sets produced similar results across most of the epidemic. However, the major difference is the daily volatility in the CQC-derived CFR. This is caused by the CQC death time series using date of report rather than date of death, which leads to strong weekend effects in the data. Additionally, as prevalence dropped during January 2021, the trends started to diverge. As prevalence started to rise from May 2021, the CFRs started to converge again. This is likely to be caused by the assumed delay distributions struggling to capture the long tail of the delay from death to death-reporting, which is most apparent after a peak, since prevalence is dropping rapidly. However, in the absence of data on the shape of this delay, it is hard to adjust for. Therefore, when possible, date of death data are more reliable for estimating the instantaneous CFR.

Across all methods and data sources, we see that the early estimates for the CFR are much larger than the later estimates. This can partially be explained by the relative lack of available testing early in the pandemic, but it could also be the case that many of the frailest care home residents died during the first wave and that treatments have improved over time [28, 29]. We also notice that during the summer months of 2020, when prevalence of the disease in England was particularly low, that the estimates of the CFR decreased, but the uncertainty increased significantly in both the forward and backward methods, because of the small numbers. To some extent, we see that the CFR curve correlates with the total cases and total deaths curves. In particular, the peaks of each curve (although less pronounced for the CFR) occur concurrently. The main driver in this case appears to be different outbreak timings across spatial scales. The relationship between prevalence and CFR is not as strong in the regional CFR estimates. Instead, when outbreaks start to take off in each region, we notice the CFR shooting up from the low summer levels to higher values. This is expected, since due to the declining epidemic, most individuals were likely to be at the tail end of their infectious period, and therefore unlikely to die [24], driving down the CFR. The mass testing programme that began over the summer will have exacerbated this fundamental bias in the CFR. As outbreaks started taking off, new infections appeared, causing the CFR to grow again [24]. This growing CFR could also have been caused by the gradually increasing frailty of care home residents, after the most frail residents died during the first wave, and by the reduced impact of false positives when prevalence is high [25, 26]. The peak CFR in the second wave is reached in December, with the national estimate around 30%, showing the high disease severity experienced by care home residents.

After January 2021, the CFR started to rapidly decline in both the regional and national analyses. This decline is mostly driven by the vaccination rollout. The vaccination prioritised care home residents initially, which led to the rapid reduction in mortality risk in this cohort. Although we can see a clear vaccine effect on the CFR, we cannot directly measure the impact of the vaccination. This is because, with cases coming down due to the national lockdown on 5th January 2021, the other factors that led to the gradual increase in CFR from September

2020 could be contributing to the decline. From March 2021, the CFR started to plateau at around 11%, and has remained constant since. This shows the significant impact of the vaccine programme. It is important to note that this 11% mortality rate is using the PHE 28-day death definition. This includes all mortality within 28 days of a positive death, and therefore will include the high baseline mortality in this cohort, where life expectancy is normally between 1–2 years [30]. Separating this baseline mortality from COVID-19 mortality is beyond the scope of this paper.

In addition to explaining the temporal trend in the English CFR, region-specific estimates are vital for understanding local risk, since demographics are not consistent across regions and have a clear impact on both the prevalence and mortality of the disease. This regional variation is apparent in the data. Using the backward method at weekly intervals, we have shown significant variation in CFR curves by region over both data sources. Specifically, we see that the CFR for London generally tracks below the national curve (after the first wave, during which CFR was unreliable due to low testing rates), whereas the North-East tends to track above, particularly in the CQC data.

Care homes may vary in the type of care provided and in the age of residents. Therefore, we estimated the CFR in different age groups of care home residents. We found that the 65–74 age group had the lowest CFR, followed by the 75–84 age groups, with the 85+ age group having the highest CFR. This is consistent with the general understanding of increased severity with age and the increased prevalence of multi-morbidity and frailty, which makes this group especially susceptible to death with COVID-19. Therefore, some of the regional differences in CFR could easily be driven by different age distributions of the local residents. To investigate type of care, we stratified care homes into whether they include nursing care. We found some evidence of increased CFR in care homes with nursing than without, as we might expect due to the increased prevalence of frailty and multi-morbidity in residents. However, the increase was small, suggesting that the main risk factor in COVID-19 mortality for care home residents is age. Note that the type of care analysis did not adjust for age. If the age distribution of residents varies significantly between care homes with and without nursing, type of care could have a different effect.

We have also demonstrated that the overall CFR is insufficient to capture the full picture of the mortality of COVID-19 in care homes in England. The temporal variation in the CFR is significant, so use of the overall CFR can be considered biased and/or outdated. The significant temporal variation means that any single value for the CFR may be unreliable when interpreting risk and, in particular, when forecasting how cases will transform into deaths, which is important for managing outbreaks. Therefore, the instantaneous CFR is necessary for understanding outbreaks in real time.

## Supporting information

**S1 Text. Additional details describing the data sources.**
(PDF)

**S2 Text. Details on the delay distribution estimation methods and results.**
(PDF)

**S3 Text. Derivation of the backward CFR methods.**
(PDF)

**S4 Text. Additional details describing the risk ratio and relative error methods.**
(PDF)

**S5 Text. Results for the risk ratio and relative error methods.**
(PDF)

## Author Contributions

**Conceptualization:** Christopher E. Overton, Luke Webb, Steve Willner.

**Data curation:** Christopher E. Overton, Uma Datta, Mike Fursman, Jo Hardstaff, Iina Hiironen, Karthik Paranthaman, James Sedgwick, Ian Hall.

**Formal analysis:** Christopher E. Overton, Luke Webb.

**Methodology:** Christopher E. Overton, Luke Webb, Lorenzo Pellis, Ian Hall.

**Software:** Christopher E. Overton, Luke Webb.

**Visualization:** Christopher E. Overton.

**Writing – original draft:** Christopher E. Overton, Luke Webb, Heather Riley, Ian Hall.

**Writing – review & editing:** Christopher E. Overton, Luke Webb, Jo Hardstaff, Karthik Paranthaman, Heather Riley, James Sedgwick, Julia Verne, Lorenzo Pellis, Ian Hall.

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
