## [Decision Letter · Decision Letter 0]

17 Feb 2022

Dear Dr Overton,

Thank you very much for submitting your manuscript "Novel methods for estimating the instantaneous and overall COVID-19 case fatality risk among care home residents in England" for consideration at PLOS Computational Biology.

As with all papers reviewed by the journal, your manuscript was reviewed by members of the editorial board and by several independent reviewers. In light of the reviews (below this email), we would like to invite the resubmission of a significantly-revised version that takes into account the reviewers' comments.

We cannot make any decision about publication until we have seen the revised manuscript and your response to the reviewers' comments. Your revised manuscript is also likely to be sent to reviewers for further evaluation.

Sincerely,

Roger Dimitri Kouyos

Associate Editor

PLOS Computational Biology

Tom Britton

Deputy Editor

PLOS Computational Biology

Reviewer's Responses to Questions

**Comments to the Authors:**

Reviewer #1: This is a well-written paper describing methods of estimating CFR using either cohort or unlinked time-series datasets. It is in very good shape and could be published as is. However, I do wonder, given that CQC data uses date of report and PHE data uses date of death, is there any benefit to using one rather than the other?

Reviewer #2: Dr. Overton and colleagues present a thorough analysis of the CFR of covid-19 among care home residents in England until October 2021. The research question is clear and the introduction provides a good overview of the question. The methods are solid and well explained, and the application is careful and appropriate. The novelty of the approach is not entirely clear, as methods similar to the presented “forward method” have been used in various situations, but the paper provides an interesting application and comparison of different approaches. I also commend the authors for their concern about uncertainty quantification. The main weakness of the study is that all comparisons (between the different methods, between age groups, between data sources, between types of care) are only done visually. The authors need to provide a quantification of these differences (with uncertainty) to support their conclusions (e.g. accuracy, sharpness, risk ratios...). There is also a lack of structure in some parts of the results, and interpretations of results need to be moved to the discussion.

Major issues

- The agreement between the different methods is only assessed visually. The authors should provide a quantification of the performance of each method (point estimates and uncertainty) compared to the chosen reference. Various metrics exist, one suggestion could be using accuracy and sharpness as proposed in

Gneiting T, Balabdaoui F, Raftery AE. Probabilistic forecasts, calibration and sharpness. Journal of the Royal Statistical Society: Series B (Statistical Methodology). 2007;69(2):243–268.

The same issue appears in comparison of CFR across time, space, age groups, and types of care. Visual comparisons are not sufficient, and the authors should provide a quantification of the differences (e.g. using risk ratios with uncertainty compared to a reference group).

- Many of the points discussed in the results have more their place in the discussion, e.g. interpretation of why the CFR was low during the summer 2020 and then increased again in section 4.2.1. The results part could also benefit from more numbered results (e.g. the CFR dropped to around 10% during the summer 2020 before increasing to about 20%…). There are also some repetitions. I feel that the results part would benefit from some level of rewriting, increasing the structure and clarity of the message.

Minor points

- A description of the characteristics of the people testing positive in UK care homes that were considered in the paper would be useful, at least regarding age and gender.

- The list of references on methods to estimate the CFR and IFR is relatively light. In particular, I saw no explicit mention of the term “preferential ascertainment of severe cases” that is one of the major biases in CFR estimates (although the author mention this bias implicitely). I would suggest that the authors consider discussing a few additional papers such as

Lipsitch, Marc, et al. "Potential biases in estimating absolute and relative case-fatality risks during outbreaks." PLoS neglected tropical diseases 9.7 (2015): e0003846.

Battegay, Manuel, et al. "2019-novel Coronavirus (2019-nCoV): estimating the case fatality rate–a word of caution." Swiss medical weekly 150.0506 (2020).

Hauser, Anthony, et al. "Estimation of SARS-CoV-2 mortality during the early stages of an epidemic: A modeling study in Hubei, China, and six regions in Europe." PLoS medicine 17.7 (2020): e1003189.

- The authors make a very good point about the fact that the CFR may decrease in time because of improvements in care and reduced virulence of variants. I would suggest adding some references on these points.

- Discussing the instantaneous CFR, the authors state that a daily resolution is too small and results in noisy estimates, and argue in favour of a weekly resolution. I am not entirely convinced by this conclusion, as this entirely depends on the incidence of infection in the population of interest. A weekly resolution may also be too noisy in situations of low incidence or in small populations, and the solution cannot always be aggregating over longer periods of time. A more general solution would be to use multi-level approaches, partially pooling information across time and space to estimate the commonalities and variation of the CFR. Other techniques such as splines or LOESS may also be useful. The authors may consider that this is out of the scope of the curent article, but should at least discuss these points as potential future improvements.

**Have the authors made all data and (if applicable) computational code underlying the findings in their manuscript fully available?**

Reviewer #1: **No: **Github is empty

Reviewer #2: Yes

PLOS authors have the option to publish the peer review history of their article (what does this mean?). If published, this will include your full peer review and any attached files.

Reviewer #1: No

Reviewer #2: No
---

## [Decision Letter · Decision Letter 1]

8 Sep 2022

Dear Dr Overton,

We are pleased to inform you that your manuscript 'Novel methods for estimating the instantaneous and overall COVID-19 case fatality risk among care home residents in England' has been provisionally accepted for publication in PLOS Computational Biology.

Best regards,

Roger Dimitri Kouyos

Academic Editor

PLOS Computational Biology

Tom Britton

Section Editor

PLOS Computational Biology

Reviewer's Responses to Questions

**Comments to the Authors:**

Reviewer #2: I am satisfied by the modifications made by the authors.

**Have the authors made all data and (if applicable) computational code underlying the findings in their manuscript fully available?**

Reviewer #2: None

PLOS authors have the option to publish the peer review history of their article (what does this mean?). If published, this will include your full peer review and any attached files.

Reviewer #2: No

---

## [Editor Report · Acceptance letter]

10 Oct 2022

PCOMPBIOL-D-21-02189R1 

Novel methods for estimating the instantaneous and overall COVID-19 case fatality risk among care home residents in England

Dear Dr Overton,

I am pleased to inform you that your manuscript has been formally accepted for publication in PLOS Computational Biology. Your manuscript is now with our production department and you will be notified of the publication date in due course.

With kind regards,

Olena Szabo
